# EFFICIENT DATA PRUNING METHODS FOR REMOTE SENSING GENERATIVE FOUNDATION MODELS

## ABSTRACT

Large-scale datasets have propelled progress in generation foundation models for remote sensing, but training on such data incurs substantial storage and compute costs. In addition, globally collected raw data often exhibit redundancy, noise, and class imbalance, which undermines training efficiency and generation quality. Existing Remote Sensing generative foundation models typically aggregate multiple classification datasets or apply simplistic deduplication, thereby overlooking the distributional requirements of generation modeling as well as the inherent heterogeneity and diversity of remote sensing imagery. To address these limitations, we propose an efficient, two-stage data pruning approach for remote sensing generative foundation models. This approach simultaneously incorporates local information content with global scene-level diversity and representativeness. Specifically, an entropy-based criterion is applied initially to efficiently eliminate low-information samples. Leveraging remote sensing scene classification datasets as reference benchmarks, we then perform scene-aware clustering with stratified sampling, which enhances the effectiveness of clustering while reducing the computational cost of clustering on large-scale unlabeled data. Finally, by balancing cluster-level uniformity with sample representativeness, the method enables fine-grained selection under high pruning ratios while preserving overall diversity and representativeness. Experiments on both curated remote sensing datasets and large-scale global data demonstrate that our pruning strategy significantly improves convergence and generation quality. Moreover, generation foundation models trained with our method consistently achieve state-of-the-art performance across multiple downstream tasks, including super-resolution and semantic image synthesis. This data pruning paradigm provides practical guidance and empirical reference for the development of remote sensing generative foundation models.

## 1 INTRODUCTION

In recent years, generation models, especially diffusion models (Peebles & Xie, 2022; Ho et al., 2020), have achieved remarkable progress in fields such as computer vision (Richard et al., 2021), medical imaging (Song et al., 2021), and remote sensing (RS) (Dong et al., 2024). Within the RS domain, generation models have been widely applied to data augmentation, image reconstruction, super-resolution, and high-resolution image synthesis, supporting practical applications in urban planning, land-use monitoring, and disaster response (Borana & Yadav, 2023). A powerful RS generative foundation model can provide a robust data and modeling backbone to further enhance these applications.

However, the effective training of RS generative foundation models critically depends on the quality and distribution of training data. The emergence of large-scale open-source datasets (e.g., Git-10M (Liu et al., 2025), RS5M (Zhang et al., 2024c)) provides valuable resources, yet it also introduces several critical challenges, including image redundancy, low-quality samples (e.g., noise and cloud cover), class imbalance (Cheng et al., 2017), and scene homogeneity (Xia et al., 2017). These issues not only hinder the training efficiency of RS foundation models but also limit the effectiveness of the resulting pretrained models on downstream tasks.

Recently, several studies in RS generation tasks have made preliminary explorations into data processing. For instance, RSDiff employs size cropping and noise augmentation (Sebaq & ElHelw,

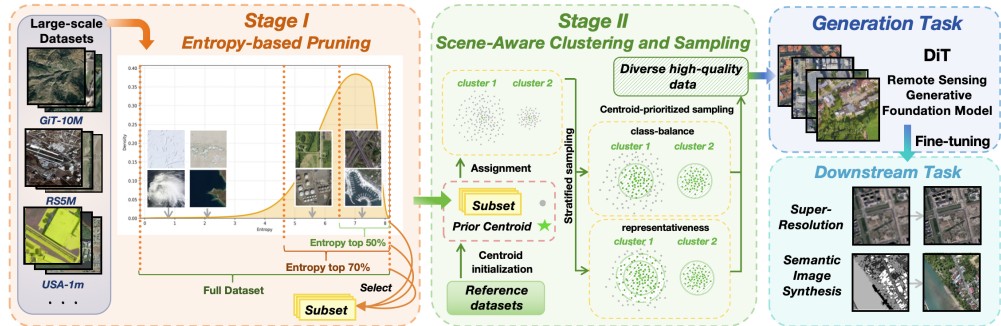

Figure 1: Overview of our multi-stage data pruning method for Remote Sensing generative foundation models.

2024). WHU-RS19 ABZSL (Balestra et al., 2025) removes cloud and open-ocean regions to reduce low information content. Super-resolution works (Huang et al., 2015) often prioritize urban areas. Nevertheless, these methods mainly rely on rule-based or simplistic strategies to eliminate redundant data, without considering the specific dependence of generation models on data distribution (Goodfellow et al., 2014) or the inherent characteristics of RS imagery, such as diversity, heterogeneity, and class balance. Moreover, many data pruning methods developed in the computer vision domain rely on scoring mechanisms (Yang et al., 2024) from supervised pretrained models (Pleiss et al., 2020). In contrast, a standardized labeled dataset for RS, comparable to ImageNet (Deng et al., 2009) in computer vision and spanning multiple resolutions and modalities, is currently lacking. It makes such scoring methods ineffective and prevents their direct application. Overall, systematic research on data selection for RS generative foundation models remains scarce, with existing approaches largely limited to basic preprocessing and lacking specialized pipelines tailored to RS generation modeling.

The effectiveness of generation foundation models depends more on the quality and distribution of the training data than on dataset size alone. In the RS domain, raw data often contains substantial redundancy and noise, meaning that merely increasing the dataset size does not guarantee proportional performance gains. Recent studies (Briq et al., 2024) have shown that diffusion models retain strong generation performance even when a large fraction (e.g., 90% of ImageNet samples) of the training data is removed, highlighting that a significant portion of the data contributes minimally. Consequently, generation models are typically trained on large volumes of data that contain considerable redundancy. Without appropriate selection, such low quality and redundant data not only slow convergence and increase computational and time costs, but may also introduce distribution shifts that degrade model performance.

To address these issues, we propose an efficient data pruning approach for RS generative foundation models. Targeting redundancy, low quality samples, and class imbalance, we systematically explore data pruning strategies across both global-scale scenarios (GiT-10M, RS5M) and urban-scale settings (our constructed USA-1m multispectral dataset). The method proceeds along two complementary dimensions: 1) Information dimension. We employ entropy-based pruning to rapidly discard low texture or homogeneous large-area regions, thereby reducing redundancy and compressing subsequent computation costs. 2) Diversity and representativeness dimension. We introduce scene–aware clustering with stratified sampling. Using existing RS scene-classified datasets as reference, we perform over-clustering on the standard datasets to obtain hundreds of cluster centroids, effectively avoiding the high computational cost of full-scale clustering. Large-scale unlabeled datasets are then assigned to these predefined clusters in the clustering space. Samples are subsequently selected using a combination of class-balanced allocation and centroid-prioritized sampling, which preferentially chooses samples near cluster centers to preserve scene-representative characteristics while maintaining diversity.

Extensive experiments reveal three key observations: First, both datasets specifically constructed for RS generation models and globally collected datasets contain substantial redundancy, and appropriate data pruning accelerates convergence while yielding models that outperform those trained on the full datasets. Second, entropy-based pruning consistently removes low quality and highly homogeneous samples, providing stable improvements across varying pruning rates. Third, optimal

pruned subsets are obtained by combining entropy-based filtering with strategies that preserve diversity and representativeness at the scene level, ensuring that informative and representative samples are retained for effective model training.

Our main contributions in this paper can be summarized as follows:

- We systematically explore data pruning for RS generativ e foundation models and propose a two-stage selection strategy that considers data heterogeneity, diversity, and representativeness, enabling faster convergence and improved generation performance.

- We introduce a reference dataset–guided clustering method, performing pre-clustering on curated scene-classified datasets to preserve the diversity of scene cluster centers while avoiding the computational cost of clustering massive unlabeled datasets.

- Extensive experiments show that our approach consistently outperforms prior state-of-the-art methods in both pretraining generation quality and downstream tasks, achieving substantial gains even with a significantly reduced training set.

## 2 RELATED WORK

### 2.1 GENERATION FOUNDATION MODELS

Generation foundation models (Saharia et al., 2022c) play a pivotal role in image generation, data synthesis, and image reconstruction. For instance, Stable Diffusion (SD) 1.5 (Rombach et al., 2022) leverages latent-space representations together with a UNet backbone, serving as a lightweight generation foundation model that provides strong pretrained initialization for ControlNet (Zhang et al., 2023) and related low-level vision tasks (Saharia et al., 2022a). More recently, Transformer-based architectures such as DiT (Peebles & Xie, 2022) have emerged, while models like SD3 (Esser et al., 2024) and Flux (Black Forest Labs, 2024) offer substantially stronger generation capabilities and higher fidelity, further advancing the scalability and versatility of diffusion frameworks.

In the RS domain, generation foundation models have also started to gain attention. To address the unique properties of RS imagery, such as multi-spectral and multi-resolution observations, geospatial information, and global coverage, many works have explored pre-training or fine-tuning foundation models on RS datasets (Tang et al., 2024; Toker et al., 2024; Xiao et al., 2023). For example, DiffusionSat (Khanna et al., 2023) introduced geolocation as conditioning information and fine-tuned Stable Diffusion 1.5 on multi-source RS data, supporting multiple RS generation tasks. Meanwhile, models such as SR3 (Saharia et al., 2022b) and CDM Ho et al. (2021), demonstrated strong super-resolution performance on natural images, providing a foundation for their adaptation to RS. Despite these advances, existing RS generation models typically aggregate multiple classification datasets or apply simplistic deduplication (Liu et al., 2024), thereby overlooking the distributional requirements of generation foundation modeling as well as the inherent heterogeneity and diversity of RS imagery. Consequently, the field lacks systematic exploration of data pruning strategies specifically designed to address the characteristics of remote sensing data and the requirements of RS generative foundation models.

### 2.2 DATA PRUNING METHODS

Training data are critical to constructing RS generative foundation models. However, existing studies (Liu et al., 2023; Sebaq & ElHelw, 2024) in this field have adopted simplistic deduplication and preprocessing strategies, such as removing cloud and ocean regions (Balestra et al., 2025) and prioritizing urban areas (Zhang et al., 2024a). These approaches are insufficient to address the sensitivity of generation models to data distribution, heterogeneity, and class imbalance.

Current data pruning methods can be broadly categorized into three types. First, data-valuation methods assign an importance score to each sample and select samples accordingly. For example, MoSo (Tan et al., 2023) estimates the change in empirical risk when a sample is removed. Although such methods are generally efficient, their performance can be affected by group effects and may lack generalization in complex real-world settings. Second, distribution-based methods rely on the geometric structure of the dataset. For instance, Moderate-DS (Xia et al., 2023b) selected samples near the median. CCS (Zheng et al., 2022) balanced data distribution and sample importance

during selection. Finally, optimization-based methods leverage optimization techniques to guide sample pruning, such as temporal dual-depth scoring (Zhang et al., 2024b), gradient matching (Killamsetty et al., 2021), scalable self-supervised pruning metrics (Sorscher et al., 2022), influence functions (Koh & Liang, 2017), and bilevel optimization (Borsos et al., 2020).

Most of these approaches are designed for supervised datasets and rely on scores generated by pre-trained supervised models. However, labeled RS data are often scarce, limiting their applicability. To bridge this gap, this study systematically explores and compares unsupervised data pruning strategies tailored to the characteristics and requirements of RS generative foundation models.

# 3 METHOD

## 3.1 WORKFLOW

Remote Sensing generative foundation models depend on large-scale, globally collected datasets that should provide high quality, diversity, and representativeness. Existing RS datasets, however, often lack these characteristics, leading to slower model convergence, suboptimal generation performance, and insufficient capability to support various low-level downstream tasks. To address this, we propose a two-stage data pruning method (Figure 1) guided by two key principles:

**1) Informational value** Cloud-covered or excessively homogeneous RS images are inevitable in global data collection. Such images are typically low-quality, contain limited informative content, and exhibit substantial redundancy. Images with higher information content, capturing meaningful structures and fine details, are prioritized for selection. This process effectively removes low-information and trivially homogeneous scenes, such as vast desert expanses, thereby preserving heterogeneity in the selected subset.

**2) Scene diversity and centroid representativeness** While maintaining the overall semantic distribution, rare scenes are preferentially preserved. For scenes with abundant samples, candidates are ranked by their similarity to the cluster centroid, and the nearest-centroid samples are selected. This strategy not only ensures high quality and de-redundancy but also aligns with expert priors reflected in reference datasets, producing a subset that is both distributionally representative and diverse.

## 3.2 STAGE I: ENTROPY-BASED PRUNING

We measure the information value of each image by computing its global Shannon entropy (Shannon, 1948). Specifically, for an image $I$, we compute its grayscale entropy $H(I)$ to capture the diversity of pixel intensities:

$$H(I) = -\sum_{k=0}^{L-1} p_k \log p_k, \tag{1}$$

where $p_k$ denotes the empirical probability of intensity level $k$ among $L$ possible levels. Images with $H(I) < \tau$ are discarded, as they typically correspond to invalid regions (e.g., sensor noise) or low-variation scenes (e.g., clouds, open ocean, deserts, or saturated exposures). This pruning step substantially reduces dataset size while preserving high-information candidate samples.

## 3.3 STAGE II: SCENE-AWARE CLUSTERING AND SAMPLING

The remote sensing domain lacks a universally adopted, comprehensive benchmark comparable to ImageNet in the natural image domain. To address this, we leverage multiple expert-curated RS classification datasets as a composable bank of clustering priors, covering various scene types such as urban areas, cropland, water bodies, forests, and transportation infrastructure. Unlike approaches that apply label-free clustering and sampling directly on a massive generic corpus, we first establish stable scene centroids on this prior bank. Importantly, instead of computing centroids separately for each labeled category, we conduct over-clustering across the entire prior dataset to obtain diverse and representative centroids. Subsequently, for the large-scale unlabeled RS dataset, samples are aligned to these centroids and selected according to centroid-prioritized sampling, ensuring both representativeness and diversity. This pipeline uses expert-curated classification datasets to derive

more representative cluster centroids, thereby enhancing diversity and representativeness in the selected subset, while simultaneously reducing the computational cost of clustering the full unlabeled large-scale dataset.

**Prior Centroid Construction.** We employ a unified feature extractor $f(\cdot)$, namely Git-RSCLIP Liu et al. (2025), with $\ell_2$ normalization to embed all images from standard datasets.Git-RSCLIP is specifically pretrained on large-scale remote sensing corpora. Compared with generic backbones such as DINOv2, it delivers more reliable and discriminative embeddings for RS imagery.

$$\mathbf{z}_x = \frac{f(x)}{\|f(x)\|_2} \quad \text{for each } x \in \mathcal{D}_{\text{ref}}, \tag{2}$$

where $\mathcal{D}_{\text{ref}}$ is the collection of reference (prior) datasets. $x$ is an image sample of the datasets. $f(\cdot)$ is a pretrained image encoder. $\mathbf{z}_x$ is the $\ell_2$-normalized embedding of $x$. We then perform $K$-means clustering on the unit hypersphere to identify representative scene centroids. Let $\mathcal{M}$ denote the set of learned centroids:

$$\mathcal{M} = \{\boldsymbol{\mu}_k\}_{k=1}^K, \quad \|\boldsymbol{\mu}_k\|_2 = 1, \tag{3}$$

where $K$ is the number of clusters and $\boldsymbol{\mu}_k$ is the $k$-th centroid in the feature space.

**Cluster Assignment and Candidate Pooling.** For each unlabeled sample $x \in \mathcal{D}_{\text{u}}$, we compute its cosine similarity to all prior centroids:

$$s_k(x) = \langle f(x), \boldsymbol{\mu}_k \rangle, \qquad k = 1, \ldots, K, \tag{4}$$

where $s_k(x)$ is the similarity score. $f(x)$ denotes the feature embedding of $x$. Each sample is assigned to the cluster with the highest similarity:

$$\hat{z}(x) = \arg\max_k s_k(x), \tag{5}$$

where $\hat{z}(x)$ is the hard cluster label. The sample is then added into the candidate pool corresponding to its assigned cluster, denoted $P_k$. This assignment procedure scales linearly with both the number of unlabeled samples and the number of centroids, making it efficient for large-scale datasets.

**Cluster-Aware Stratified Sampling.** Given a total sampling budget $B$, we combine class-balanced allocation with centroid-prioritized sampling.

- **Class-balanced allocation.** Let $\{P_k\}_{k=1}^K$ denote the candidate pools for $K$ clusters, and let $q = \lfloor B/K \rfloor$. We assign each cluster a quota of $q$ samples to ensure coverage across clusters. Rare clusters with fewer than $q$ samples retain all candidates, preserving diversity.

- **Centroid-prioritized sampling.** Within each cluster $k$, we rank samples $x \in P_k$ by their similarity $s_k(x)$ to the cluster centroid $\boldsymbol{\mu}_k$ and select the top-$q$ samples:

$$S_k = \underset{x \in P_k}{\text{Top-q}} \; s_k(x). \tag{6}$$

If $|P_k| < q$, we set $S_k = P_k$ and reallocate the remaining budget by selecting additional samples from the global remainder: $P_{\text{rem}} = \bigcup_{j=1}^K (P_j \setminus S_j)$ in descending order of similarity until the total number of selected samples equals $B$, i.e., $B = \sum_{k=1}^K |S_k|$.

## 3.4 COMPLEXITY ANALYSIS

In stage I, the overall complexity is approximately $\mathcal{O}(N)$, as each image is evaluated individually for information value.

In the stage II, we have: 1) Building clustering priors on a small, standardized reference set is done once, so its cost is negligible relative to the full pipeline. 2) Assigning each image to the most similar centroid requires computing similarities to $K$ centroids in a $f_d$-dimensional feature space, giving

$\mathcal{O}(NKf_d)$. 3) Within each cluster, ranking images by similarity and sampling $q$ samples requires sorting, which has a complexity of $\mathcal{O}\left(\frac{N}{K}\log\frac{N}{K}\right)$ per cluster, resulting in $\mathcal{O}\left(N\log\frac{N}{K}\right)$. Since $K$ and $f_d$ are small constants in practice, the end-to-end complexity can be considered $\mathcal{O}(N\log N)$. The method does not involve training deep models. The core operations are vector arithmetic, similarity computation, and sorting. Consequently, the pipeline is highly scalable and practically deployable for data pruning over large-scale RS datasets.

## 4 EXPERIMENTS

### 4.1 DATASETS AND EVALUATION

To evaluate the effectiveness of our data pruning approach for RS generation tasks, we conduct experiments on both global-scale and urban-scale datasets. Specifically, we use two representative global-scale optical datasets (i.e., GIT-10M and RS5M) that were carefully curated for RS generative foundation models, as well as a large-scale urban-scale multispectral dataset (i.e., USA-1m) collected in this work based on U.S. urban boundary products (Li et al., 2020). The global datasets provide broad coverage and diversity, while the urban dataset offers finer spatial resolution and richer texture information. This design allows us to assess the effectiveness and generalizability of our data pruning strategy across datasets with varying resolutions, modalities, volumes, and sensors.

- **Git-10M** (Liu et al., 2025) a *global-scale* dataset comprising 10.5M satellite images (RGB) with a spatial resolution of 0.5–128 m, which spans multiple continents and geographic regions, covering diverse land-cover types such as urban areas, croplands, forests, mountains, and deserts.

- **RS5M** (Zhang et al., 2024c) a *global-scale* dataset initially collected from 11 publicly available image–text pair datasets, containing RS images (RGB) from different regions, resolutions, and scene types. Due to large variations in image size and relatively low data quality, we apply simple preprocessing and filtering, resulting in a curated subset of about 1.04M images.

- **USA-1m (multispectral)** our self-constructed, urban-scale remote sensing dataset consisting of 8.77M four-channel, high-resolution (1 m) images, derived from the USDA National Agriculture Imagery Program (NAIP) (Robinson et al., 2019). The dataset covers major urban areas across 45 U.S. states, with a total footprint of $1.34339 \times 10^6$ km$^2$. A visualization of the spatial distribution of the dataset is provided in Figure 4 (see Appendix F).

For each of the aforementioned datasets, we selected 5,000 high-quality images for subsequent metric computation for the generation task.

In Stage II, we adopt five scene-classification datasets with distinct characteristics: NWPU-RESISC45 (Cheng et al., 2017), UC Merced Land-Use (Yang & Newsam, 2010), AID (Xia et al., 2017), WHU-RS19 (Balestra et al., 2025), and RSD46-WHU (Long et al., 2017). Detailed information regarding their resolution, dataset size, number of classes, and data sources can be found in Appendix F.

To comprehensively assess the impact of training data on RS generation models, we evaluate their generation capability directly on the test set and further examine their effectiveness as pretrained models on two downstream tasks, i.e., super-resolution (SR) and semantic image synthesis (SIS). The evaluation metrics include FID (Heusel et al., 2017) and LPIPS (Zhang et al., 2018). Lower LPIPS and FID values indicate superior generation performance.

- **SECOND-SR** we construct a super-resolution dataset based on the high-resolution SECOND semantic change detection benchmark (Yang et al., 2020). Low-resolution (LR) images are generated by first downsampling and then upsampling the original high-resolution (HR) images. The dataset contains 4,662 paired LR–HR images, each with a spatial size of $512\times512$ pixels, covering six major land-cover classes.

- **OpenEarthMap** (Xia et al., 2023a) a global benchmark designed for semantic segmentation. It consists of 5,000 RS images across six continents at 0.25 to 0.5 m resolutions. Labels are annotated with eight land-cover categories.

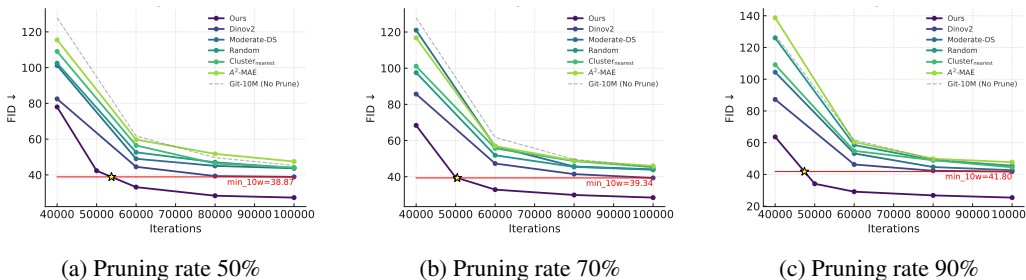

(a) Pruning rate 50%        (b) Pruning rate 70%        (c) Pruning rate 90%

Figure 2: Comparison of generation performance (FID) across different pruning rate on Git-10M.

## 4.2 IMPLEMENTATION DETAILS

All models are implemented using the official DiT framework (Peebles & Xie, 2022),which is a transformer-based architecture using its github repository[1]. We adopt the DiT-XL/2 backbone for both generation pretraining and downstream tasks. All experiments are conducted with $256 \times 256$ input size, a global batch size of 256, and AdamW optimizer with a fixed learning rate of $1 \times 10^{-4}$. Training is distributed across 4 NVIDIA H100 GPUs. **Base VAE** For Git-10M and RS5M, we use the publicly released Stable Diffusion VAE (`sd-vae-ft-ema`)[2]. For USA-1M, we train a custom four-channel VAE from scratch and select the checkpoint with the lowest validation loss. **Generation pretraining** DiT is trained in an unconditional setting. For each dataset, we run 40K to 100K diffusion steps and models are trained from scratch. **Downstream fine-tuning** For super-resolution and semantic image synthesis, we initialize from the pretrained DiT checkpoints and fine-tune for 5,000 steps per task.

## 4.3 COMPARISON RESULTS

We compare the proposed method with three groups of pruning strategies: basic pruning methods (e.g., Random, Moderate-DS (Xia et al., 2023b)), data pruning methods for foundation models (e.g., Dinov2 (Oquab et al., 2023), $A^2$-MAE (Zhang et al., 2024a)), and a recent approach specifically developed for generation models (Cluster$_{nearest}$ (Briq et al., 2024)). To examine their effectiveness across varying scene complexities, experiments are conducted on a global-scale dataset (Git-10M) and an urban-scale dataset (USA-1m). To ensure comparability, all models are trained for an equivalent number of iterations, 100K for Git-10M and 40K for USA-1m, and the best results during training are reported. Beyond generation evaluation on the test set, pretrained models are further assessed on two downstream tasks, i.e., SR and SIS, with the top fine-tuning results recorded.

As illustrated in Figure 2, using Git-10M as an example, our method converges faster than the comparison methods across different pruning ratios. Table 1 presents results at a fixed pruning ratio of 70% on both datasets. The proposed method achieves the best performance across generation and downstream tasks, demonstrating strong robustness and generalization. In addition, $A^2$-MAE, designed for global-scale datasets, exhibits a notable performance drop on the urban-scale USA-1m. On Git-10M, our method achieves the lowest generation FID (28.45), representing a 27.7% improvement over the next best score (39.34). It further improves performance on super-resolution and semantic image synthesis, demonstrating strong alignment between downstream metrics and generation FID. On USA-1m, our approach also achieves consistent improvements across metrics, further confirming its effectiveness and generalizability to multispectral remote sensing imagery.Due to limited space, more in-depth analyses regarding training steps are provided in Appendix A.

## 4.4 ABLATION STUDY

### 4.4.1 EFFECTIVENESS OF ENTROPY-BASED PRUNING

To validate the effectiveness of entropy-based pruning, we conduct experiments on the Git-10M dataset. Images are ranked in descending order of Shannon entropy, and the top $p\%$ subset is retained, denoted as $\mathcal{D}_{p\%}^{H\uparrow}$, while the unfiltered dataset is denoted as $\mathcal{D}$all. Under a fixed training

---

[1]https://github.com/facebookresearch/DiT
[2]https://huggingface.co/stabilityai/sd-vae-ft-ema

Table 1: Experimental results with different data pruning methods on Git-10M and USA-1m. We report results at a pruning ratio of 70%.

| Method | Git-10M | | | | | USA-1m | | | | |
|---|---|---|---|---|---|---|---|---|---|---|
| | Generation | SR | | SIS | | Generation | SR | | SIS | |
| | FID↓ | FID↓ | LPIPS↓ | FID↓ | LPIPS↓ | FID↓ | FID↓ | LPIPS↓ | FID↓ | LPIPS↓ |
| Full Dataset | 45.30 | 89.25 | 0.3912 | 125.79 | 0.6027 | 242.52 | 125.90 | 0.4909 | 327.33 | 0.8212 |
| Random | 43.87 | 89.25 | 0.3912 | 127.00 | 0.6200 | 255.11 | 126.63 | 0.4827 | 236.68 | 0.6698 |
| Moderate-DS (Xia et al., 2023b) | 44.07 | 92.40 | 0.3952 | 132.85 | 0.6222 | 199.24 | 124.86 | 0.5098 | 233.13 | 0.6730 |
| Cluster$_{nearest}$ (Briq et al., 2024) | 45.24 | 89.58 | 0.3898 | 133.18 | 0.6237 | 201.36 | 150.11 | 0.5870 | 256.40 | 0.6804 |
| Dinov2 (Oquab et al., 2023) | 39.34 | 91.76 | 0.3930 | 137.55 | 0.6121 | 180.98 | 127.73 | 0.5472 | 238.80 | 0.6761 |
| A$^2$-MAE (Zhang et al., 2024a) | 45.96 | 90.39 | 0.3938 | 130.64 | 0.6013 | 360.74 | 148.36 | 0.6028 | 256.39 | 0.6993 |
| **Ours** | **28.46** | **87.98** | **0.3893** | **122.08** | **0.5967** | **175.93** | **122.00** | **0.4779** | **199.82** | **0.6682** |

Table 2: Results of entropy-based pruning at different pruning ratios on GiT-10M, reporting FID for the generation task.

| Dataset | $\mathcal{D}_{all}$ | $\mathcal{D}_{70\%}^{H\uparrow}$ | $\mathcal{D}_{50\%}^{H\uparrow}$ | $\mathcal{D}_{30\%}^{H\uparrow}$ | $\mathcal{D}_{25\%}^{H\uparrow}$ | $\mathcal{D}_{20\%}^{H\uparrow}$ | $\mathcal{D}_{15\%}^{H\uparrow}$ | $\mathcal{D}_{5\%}^{H\uparrow}$ |
|---|---|---|---|---|---|---|---|---|
| FID ↓ | 61.6887 | 40.8934 | 33.1489 | **32.8622** | 33.6968 | 34.9136 | 36.8073 | 41.8878 |

budget of 60K iterations, generation models trained on each subset are evaluated using FID (Table 2). On this global-scale dataset, retaining only the top 30% of images achieves the best FID (32.86), substantially outperforming the full dataset (61.69), highlighting the importance of removing low-information data. We further observe that as the pruning threshold becomes more stringent, FID initially decreases and then increases, indicating that the optimal pruning ratio should be chosen based on the redundancy level of the dataset. Analyses of different datasets and optimal pruning ratios are provided in Appendix D, Table 6.

### 4.4.2 EFFECTIVENESS OF SCENE−AWARE CLUSTERING AND SAMPLING

In this section, we conduct experiments on two datasets with different scales, including GiT-10M (tens of millions) and RS5M (millions), to evaluate the effectiveness of Stage II. As shown in Table 3, Stage II consistently provides stable gains under high pruning ratios, exhibiting a consistent trend across both datasets. On GiT-10M, pruning at 85% and 90% leads Stage II to outperform Stage I alone by approximately 10%. Similarly, on RS5M, Stage II delivers notable benefits under high pruning, with corresponding improvements in downstream SIS performance.

At low pruning ratios, Stage II yields minimal gains or may perform slightly worse, indicating that clustering-based selection depends on prior removal of low-quality data, otherwise balanced and scene-representative sampling can be compromised. When pruning is light and the subset remains large, entropy-based pruning alone is sufficient. Under high pruning ratios, sample representativeness becomes insufficient, and clustering is required to preserve diversity and distributional coverage. Overall, Stage II demonstrates robust and transferable effectiveness across both large- and medium-scale datasets, with especially pronounced advantages in highly compressed scenarios.And additional ablations on sampling strategies in Step II can be found in Appendix C.

### 4.4.3 EFFECTIVENESS OF REFERENCE DATASET−GUIDED CLUSTERING METHOD

We further perform ablation studies on the proposed reference dataset–guided clustering strategy in comparison with the standard practice of clustering directly on the unlabeled dataset. As reported in Table 4, our approach consistently surpasses clustering the full unlabeled collection in both runtime efficiency and generative quality. Even the widely adopted MiniBatch KMeans variant, which reduces the feature dimensionality from 1024 to 128 with subsequent normalization, remains less effective. The improvement arises from exploiting expert-curated reference datasets to derive more representative scene prototypes, thereby facilitating faster sample assignment and enhancing FID.

We also examine the influence of reference dataset selection, varying in scene diversity and dataset scale, together with the associated degree of over-clustering. Experiments conducted with datasets of different complexity levels and their combinations (Appendix B, Figure 3) yield several observations. An important finding is that the diversity of scene categories in the reference dataset plays a critical role, as datasets with broader coverage produce more representative clustering outcomes.

Table 3: Results of scene-aware clustering and sampling at different pruning ratios. The subset refers to the Stage I output used as input for Stage II.

| Pruning ratio | Subset | Stage I | Stage II | Git-10M | | | RS5M | | |
|---|---|---|---|---|---|---|---|---|---|
| | | | | generation | SIS | | generation | SIS | |
| | | | | FID↓ | FID↓ | LPIPS↓ | FID↓ | FID↓ | LPIPS↓ |
| – | $\mathcal{D}_{all}$ | | | 127.8932 | 143.2622 | 0.6236 | 113.3864 | 137.9567 | 0.6076 |
| 30% | $\mathcal{D}_{70\%}^{H\uparrow}$ | ✓ | | 100.3227 | 134.2329 | 0.6064 | 81.9246 | 122.9466 | 0.5950 |
| 50% | $\mathcal{D}_{50\%}^{H\uparrow}$ | ✓ | | 77.9903 | 127.7947 | 0.6069 | 71.6860 | 121.5837 | 0.5970 |
| | $\mathcal{D}_{70\%}^{H\uparrow}$ | ✓ | ✓ | 84.3062 | 134.8372 | 0.6172 | 80.7609 | 121.7632 | 0.6002 |
| 70% | $\mathcal{D}_{30\%}^{H\uparrow}$ | ✓ | | 68.3773 | 126.7446 | 0.6074 | 63.3343 | 125.0599 | 0.5971 |
| | $\mathcal{D}_{50\%}^{H\uparrow}$ | ✓ | ✓ | 70.0361 | 131.7129 | 0.6045 | 63.3343 | 125.0599 | 0.5950 |
| 85% | $\mathcal{D}_{15\%}^{H\uparrow}$ | ✓ | | 68.7814 | 125.7938 | 0.6044 | 59.4955 | 126.7798 | 0.5966 |
| | $\mathcal{D}_{30\%}^{H\uparrow}$ | ✓ | ✓ | **61.3269** | **120.5167** | **0.5982** | **58.9813** | **115.6430** | **0.5902** |
| | $\mathcal{D}_{50\%}^{H\uparrow}$ | ✓ | ✓ | 64.4233 | 139.9440 | 0.6139 | 73.7312 | 132.6490 | 0.6014 |
| | $\mathcal{D}_{70\%}^{H\uparrow}$ | ✓ | ✓ | 72.3580 | 130.6310 | 0.6139 | 75.4898 | 121.8777 | 0.5941 |
| 90% | $\mathcal{D}_{10\%}^{H\uparrow}$ | ✓ | | 70.9027 | 121.9285 | 0.6043 | 65.4067 | 124.6616 | 0.6034 |
| | $\mathcal{D}_{30\%}^{H\uparrow}$ | ✓ | ✓ | 63.6400 | 123.0029 | 0.6106 | 64.2536 | 124.8206 | 0.6004 |
| | $\mathcal{D}_{50\%}^{H\uparrow}$ | ✓ | ✓ | 64.2870 | 144.4404 | 0.6173 | 71.3744 | 130.5425 | 0.6094 |
| | $\mathcal{D}_{70\%}^{H\uparrow}$ | ✓ | ✓ | 77.2366 | 131.4739 | 0.6094 | 79.8913 | 128.3239 | 0.5970 |

Table 4: Generative quality and runtime of clustering on full unlabeled data, entropy-pruned unlabeled data and our reference dataset-guided approaches.

| Method | Cluster sample size | Cluster Numbers | Feature Dimension | Times(s) | FID↓ |
|---|---|---|---|---|---|
| Full unlabeled clustering | 10.5 million | 200 | 1024 | 4630.3 | 108.84 |
| Entropy-pruned unlabeled clustering | 3.1 million | 200 | 128 | 308.4 | 71.16 |
| Reference-guided (single, 21 classes) | 2,100 | 200 | 1024 | 82.3 | 61.69 |
| Reference-guided (single, 45 classes) | 31,500 | 200 | 1024 | **76.6** | **60.14** |
| Reference-guided (3 datasets) | 43,600 | 200 | 1024 | 101.8 | 62.28 |
| Reference-guided (5 datasets) | 55,605 | 200 | 1024 | 115.1 | 60.65 |

Another key observation is that the number of clusters $K$ admits a moderate optimal range, with $K=200$ emerging as a robust configuration that adequately covers common RS scenes while avoiding the breakdown of scene-balanced sampling caused by excessive fragmentation. Finally, we note that a simple aggregation of multiple scene classification datasets is not automatically beneficial, since distributional discrepancies across datasets may impede the alignment of semantically related categories and thus undermine representativeness. Therefore, effective dataset integration requires careful consideration of both scene complementarity and domain compatibility.

Due to the limited space, limitation and future work can be seen in Appendix E.

# 5 CONCLUSION

In this work, we comprehensively explore data pruning for RS generative foundation models across datasets of varying scale and coverage. We propose a two-stage pruning strategy that jointly considers data heterogeneity, diversity, and representativeness. This approach consists of informativeness-based pruning followed by scene-aware clustering with sampling, enabling the construction of a high-quality subset that is both representative and diverse. By leveraging RS scene-classified datasets as references for over-clustering, the method preserves the diversity of scene cluster centers while avoiding the computational cost of clustering massive unlabeled datasets. Extensive experiments demonstrate the effectiveness of our approach and provide practical guidance and empirical insights for the development of future Remote Sensing generative foundation models.

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

## A    EXTENDED COMPARISON UNDER DIFFERENT TRAINING ITERATIONS

We adopt an early-stopping protocol of 40K iterations for the ablations, effectiveness studies, and most comparative experiments to cover more methods and settings under a feasible compute budget. This configuration reliably captures relative rankings and trends across methods while substantially reducing the per-run cost. To verify the robustness of the 40K-iteration conclusions, we additionally train all compared methods to 100K iterations under identical data and hyperparameters, and results are shown in Figure 2.

**1) Reliability of the early proxy.** From 40K to 100K iterations, FID decreases roughly monotonically, and the relative rankings and trends at 40K closely match those at 100K, across all pruning ratios, Ours consistently attains the lowest FID. Hence, 40K serves as a reliable early proxy for model selection, preserving the final (100K) ordering while substantially reducing compute.

**2) Convergence speed and compute savings.** Ours converges markedly faster. At 60K iterations, Ours (90% pruning) reaches an FID of 29.14, outperforming the best 100K result of competing methods (38.87) by about 25%. By 80K, the margin widens to roughly 31%. Even at 50K, our FID approaches the best scores of competitors at 100K. Thus, our method can maintain strong performance using only 10% of the data while saving over 50% of the compute.

## B    EXPLORATION OF THE OPTIMAL NUMBER OF REFERENCE DATASETS
AND CLUSTERS

We examine how reference dataset selection (scene diversity and scale) and the degree of over-clustering affect outcomes (see Figure 3). Experiments across datasets of varying complexity and their combinations yield the following takeaways:

**1) A simple aggregation of multiple scene classification datasets is not automatically beneficial.** Moving from one to three sources markedly reduces FID by enriching scene coverage and improving representativeness. However, simply aggregating five sources increases volatility and does not surpass the best results with fewer references, likely due to cross-dataset heterogeneity that destabilizes centroids and misaligns semantically related categories.

**2) A moderate cluster count is optimal.** Increasing $K$ from small to moderate consistently helps, but overly large $K$ fragments scenes, dilutes centroids, and breaks scene-balanced sampling, leading to regressions. In our setting, $K=200$ emerges as a robust configuration that adequately covers common RS scenes without over-fragmentation.

**3) The diversity of scene categories in the reference dataset plays a critical role.** With a single reference, a scene-rich set with broader coverage produce more representative clustering outcomes than narrower sets under the same $K$. If such a set is unavailable, combining a small number (e.g., 2 to 3) of complementary datasets is effective, provided $K$ is tuned in the moderate range.

## C    ADDITIONAL ABLATIONS ON SAMPLING STRATEGIES IN STEP II

We fix the backbone to DiT, use the entropy-filtered subset $\mathcal{D}_{30\%}^{H\uparrow}$, and set the number of clusters to $K=70$ , pruning ratio is 85%. Each run trains for 40K steps and samples 5K images. We ablate the inter-class allocation (how the sampling budget is distributed across clusters) and the intra-class sampling (how samples are scored within a cluster). The details of strategy are as follows:

**1) Uniform quota & centroid-prioritized sampling.** Assign an equal quota $q$ to each cluster. Within each cluster, select samples with the highest cosine similarity to the centroid. If a cluster has fewer than $q$ candidates, fill the shortfall by resampling *with replacement* until the quota is met.

**2) Class-balanced allocation & farthest-from-centroid sampling.** Assign an equal quota $q$ to each cluster. Within each cluster, select the farthest-from-centroid samples to promote diversity. If a cluster has fewer than $q$ candidates, take all available; any remaining global shortfall is filled by selecting the lowest-similarity samples from the overall pool.

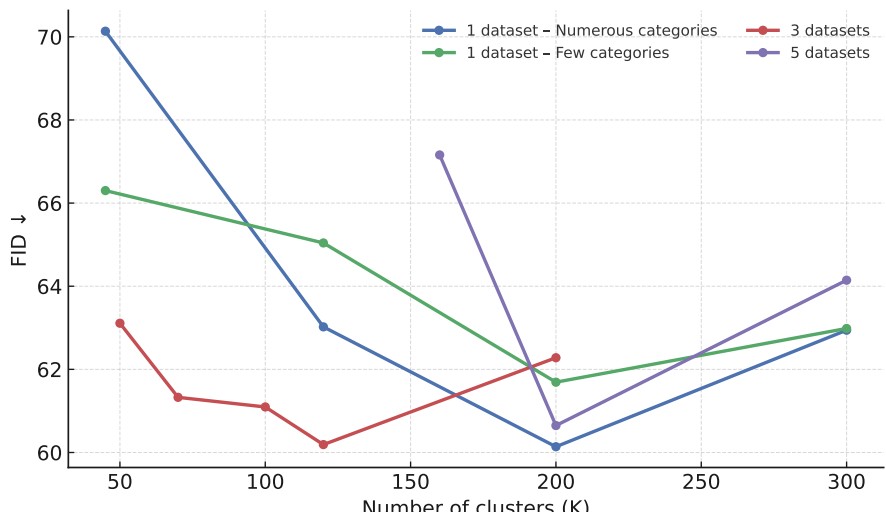

Figure 3: Effect of Reference Datasets and Cluster Count on FID.

Table 5: Ablations of step II strategies on $\mathcal{D}_{30\%}^{H\uparrow}$.

| Inter-class allocation | Intra-class sampling | FID↓ |
|---|---|---|
| Uniform quota | Nearest to centroid | 66.53 |
| class-balanced alllocation | Farthest from centroid | 69.69 |
| Size-proportional quota | Density-based sampling | 64.37 |
| Dispersion-proportional quota | Nearest to centroid | 66.97 |
| Few-shot priority | Moderate-DS sampling | 69.84 |
| Entropy-proportional quota | Nearest to centroid | 62.22 |
| Dispersion-proportional quota | Nearest to centroid | 66.42 |
| Complexity-proportional quota | Nearest to centroid | 73.61 |
| Uniform + uniform remainder | Nearest to centroid | 67.40 |
| Uniform + proportional remainder | Nearest to centroid | 64.12 |
| Uniform + LOF remainder | Nearest to centroid | 70.00 |
| **Ours** | | **61.32** |

**3) Size-proportional quota & density-based sampling.** Allocate the quota in proportion to each cluster's size (number of candidates). Within each cluster, apply a density criterion (e.g., $k$NN density) to draw more from dense regions and fewer from sparse ones.

**4) Dispersion-proportional quota & centroid-prioritized sampling.** Allocate the quota according to cluster dispersion (mean distance from samples to the centroid). Within each cluster, prefer near-centroid samples. If a cluster cannot meet its quota, backfill proportionally from other clusters without exceeding their available counts.

**5) Entropy-proportional quota & centroid-prioritized sampling.** Allocate the quota in proportion to intra-cluster entropy (spectral entropy of the feature covariance). Within each cluster, prefer near-centroid samples. If underfilled, backfill as in Item 4.

**6) Few-shot priority & Moderate-DS sampling.** First take all samples from low-cardinality (few-shot) clusters, then distribute the remaining budget in proportion to cluster size. Within each cluster, apply the Moderate-DS sampling rule (Xia et al., 2023b).

**7) LOF-proportional quota & centroid-prioritized sampling.** Allocate the budget across clusters in proportion to their LOF-based scores. Within each cluster, select samples nearest to the centroid. If a cluster cannot meet its quota, backfill from the remaining clusters.

**8) Complexity-proportional quota (mean image entropy) & centroid-prioritized sampling.** Allocate the budget in proportion to cluster complexity measured by mean image entropy. Within each cluster, select samples nearest to the centroid. If underfilled, backfill from the remaining clusters.

**9) Uniform quota with iterative uniform remainder & centroid-prioritized sampling.** Start with an equal quota $q$ per cluster. If some clusters fall short, redistribute the remaining budget uniformly over the remaining clusters in rounds. Within each cluster, select samples nearest to the centroid.

**10) Uniform quota with iterative size-proportional remainder & centroid-prioritized sampling.** Start with an equal quota per cluster. Reassign any leftover budget in proportion to residual cluster size (unused candidates). Within each cluster, select samples nearest to the centroid.

**11) Uniform quota with iterative LOF-proportional remainder & centroid-prioritized sampling.** Start with an equal quota per cluster. Allocate any leftover budget across clusters in proportion to their LOF-based scores. Within each cluster, select samples nearest to the centroid.

We systematically compared a variety of inter-class allocation and intra-class sampling combinations to identify a suitable strategy. Balancing quality, stability, and diversity, we adopt Class-balanced allocation and Centroid-prioritized sampling in Step II,which assign an equal quota per cluster to mitigate long-tail imbalance, then select within each cluster by descending similarity to the centroid to preserve representativeness and match the target distribution,finally supplemental sampling by descending similarity. This strategy attains the best FID in our ablations (61.3269) while remaining simple and low-overhead (dominated by vector similarity and sorting), showing strong robustness and scalability for large-scale remote-sensing data selection.

## D  PRUNING RATIO RECOMMENDATIONS

We use spherical K-Means ($K \approx N/200$) to analyze the cluster statistics of datasets. For RS5M dataset ,we find that the cluster utilization rate $U$ is extremely low, while the maximum-cluster ratio $h$ is extremely high (Table 6). In other words, the majority of samples are "absorbed" by a few giant city clusters. Under such a distribution, many fine-grained scenes (e.g., ports, industrial zones, coastal suburban areas) are merged into the same urban mega-cluster. Stage II, with its "center-prior" sampling, tends to favor prototype urban textures near the cluster center, while pushing away boundary or rare sub-patterns. This further suppresses the already scarce long-tail categories, leading to a loss of diversity. As a result, entropy-only pruning (i.e., Stage I) performs better in this setting.

Based on our existing experimental results, we provide empirical recommendations for the pruning ratio:

For large-scale datasets with relatively balanced distributions ($>$1M samples), a higher pruning ratio can be adopted. In contrast, for datasets with spiky long-tail distributions and fewer categories, a moderate pruning ratio is more suitable.

Under high pruning ratios, we recommend the Stage I + Stage II combination, which balances representativeness and diversity. Under low pruning ratios, entropy-only pruning (Stage I) is preferable.

## E  LIMITATION AND FUTURE WORK

Our study focuses on single-modality generation models for RS and does not investigate cross-modal relationships. On one hand, textual information in RS typically comes from land-cover products or GPT-generated captions, which are less diverse and rarely suffer from image-text misalignment compared to general-domain datasets. On the other hand, current RS field still lacks large-scale

Table 6: Dataset statistics and the empirically best pruning ratio. $U$ is cluster utilization. $h$ is max-cluster share. $R$ is redundancy.

| Dataset | Count ($\times 10^4$) | $U$ (%) | $h$ (%) | $R$ | Best ratio |
|---------|-----------------------|---------|---------|-----|------------|
| Git-10m | 1050 | 92 | 0.011 | 0.9956 | 85% |
| USA-1m | 877 | 5.4 | 2.09 | 0.9998 | 70% |
| RS5M | 143 | 89 | 0.085 | 0.9969 | 85% |

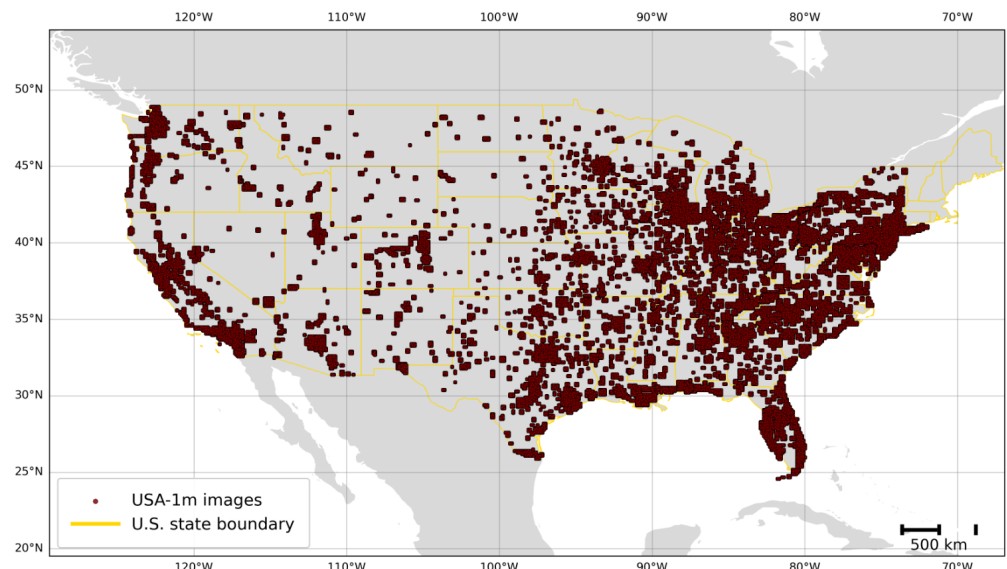

Figure 4: Visualization of the spatial distribution of USA-1m.

multi-modal generation foundation models and datasets that unify SAR, multispectral, hyperspectral, and RGB imagery. Consequently, our experiments mainly consider pruning visible and near-infrared data. Future work could explore the construction of multimodal generative datasets and the incorporation of inter-modal correlations.

In addition, we plan to investigate training remote sensing generative foundation models based on data-pruned datasets. Proper data pruning can substantially accelerate model convergence. As computational resources and multi-source datasets increase, such high-quality pruned datasets can further enhance model capacity. Through multi-stage training and the incremental incorporation of new data, data-pruned subsets can serve both to speed up convergence and reduce training difficulty, enabling the progressive improvement of the foundation model performance.

## F    DATASET

- **NWPU-RESISC45** (Cheng et al., 2017) including 31,500 RS images, each of size $256 \times 256$ pixels, covering 45 scene categories. The imagery is collected from Google Earth at 0.2 to 5 m resolutions.

- **UC Merced Land-Use** (Yang & Newsam, 2010) including 2,100 RS images , each of size $256 \times 256$ pixels, covering 21 land-use classes. The imagery is sourced from the USGS National Mapat 0.3 m resolutions.

- **AID** (Xia et al., 2017) including 10,000 RS images, each of size $600 \times 600$ pixels, covering 30 scene categories. The imagery is collected from Google Earth at 0.5 to 8 m resolutions.

- **WHU-RS19** (Balestra et al., 2025) including 1005 RS images, each of size $600 \times 600$ pixels, covering 19 scene categories. The imagery is collected from Google Earth at 0.5 m resolutions.

Table 7: Summary of datasets used in this study.

| Category | Dataset | Year | Volume | Bands | Image size | Resolution | Classes | Coverage / Scene |
|---|---|---|---|---|---|---|---|---|
| Pruning datasets | Git-10M (Liu et al., 2025) | 2025 | 10.5M | RGB | varying | 0.5–128 m | – | global |
| | RS5M (Zhang et al., 2024c) | 2024 | 1.04M | RGB | varying | varying | – | global |
| | USA-1m (Robinson et al., 2019; Li et al., 2020) | 2020 | 8.77M | 4 | 512 | 1 m | – | U.S. urban areas |
| Standard Datasets | NWPU-RESISC45 (Cheng et al., 2017) | 2017 | 31,500 | RGB | 256 | 0.2–5 m | 45 | diverse scenes |
| | UC Merced Land-Use (Yang & Newsam, 2010) | 2010 | 2,100 | RGB | 256 | 0.3 m | 21 | land-use scenes |
| | AID (Xia et al., 2017) | 2017 | 10,000 | RGB | 600 | 0.5–8 m | 30 | diverse scenes |
| | WHU-RS19 (Balestra et al., 2025) | 2025 | 1,005 | RGB | 600 | 0.5 m | 19 | high-resolution scenes |
| | RSD46-WHU (Long et al., 2017) | 2017 | 117,000 | RGB | 256 | 0.5–2 m | 46 | diverse scenes |
| SR SIS | SECOND (?) | 2021 | 4,662 | RGB | 512 | 0.5 m | 6 | change detection regions |
| | OpenEarthMap (Xia et al., 2023a) | 2023 | 5,000 | RGB | 512 | 0.25–0.5 m | 8 | global |

- **RSD46-WHU** (Long et al., 2017) including 117,000 RS images, each of size $256 \times 256$ pixels, covering 46 scene categories. The imagery is collected from Google Earth at 0.5 to 2 m resolutions.

Additionally, information on all datasets mentioned in this paper is summarized in Table 7.

# G   THE USE OF LARGE LANGUAGE MODELS

LLM was used to aid language polishing.

