# OpenReview forum: "Efficient Data Pruning Methods for Remote Sensing generative foundation Models"
_ICLR.cc/2026/Conference — ICLR 2026 Conference Withdrawn Submission_

### Official Review · Reviewer_jhCu · 2025-10-30

**Soundness:** 2
**Presentation:** 2
**Contribution:** 2
**Rating:** 2
**Confidence:** 4

**Summary:**

This paper proposes an efficient, two-stage data pruning strategy to address the high computational cost and performance issues caused by large, redundant, and low-quality remote sensing (RS) datasets used for training generative foundation models. The first stage employs an entropy-based filter to rapidly discard low-information samples like clouds, deserts, or open oceans. The second, novel scene-aware clustering and sampling stage avoids the extreme computational cost of clustering the entire massive dataset ; instead, it first generates representative scene centroids by over-clustering a "prior bank" of smaller, expert-curated RS classification datasets. The large-scale unlabeled data is then efficiently assigned to these pre-defined centroids, and a stratified, centroid-prioritized sampling method selects a final subset that balances scene diversity and representativeness. Experiments show this method significantly improves convergence and generation quality, allowing models trained on highly pruned datasets to outperform models trained on the complete datasets in both generation and downstream tasks like super-resolution.

**Strengths:**

1. The paper's most significant contribution is the "reference dataset-guided" clustering in Stage II. Instead of incurring the massive computational cost of clustering the entire multi-million-image unlabeled dataset , the method efficiently pre-computes "prior centroids" from smaller, expert-curated classification datasets. This novel approach is shown to be not only much faster but also more effective, leading to better FID scores.
2. A major practical benefit is the acceleration of model training. The paper clearly demonstrates that models trained on its pruned subset converge significantly faster than those trained on the full dataset or subsets from other methods.
3. he paper tackles a critical, practical bottleneck in the development of foundation models: the massive, noisy, and highly redundant nature of globally-collected remote sensing data. By providing an effective and efficient data pruning pipeline, the paper offers a valuable and practical solution for the field.

**Weaknesses:**

1. The conclusions drawn from the ablation studies are confusing and seem to conflict with the main results. For example, Table 3 shows that at a 70% pruning ratio Stage I-only achieves a better FID (68.38) than the full two-stage method (70.04). This leads the authors to recommend that "Under low pruning ratios, entropy-only pruning (Stage I) is preferable". However, the paper's main result in Table 1 is also at a 70% pruning ratio, where the two-stage method ("Ours") achieves the best FID.
2. There are concerning FID scores for the main method that were not addressed. Table 2 shows that stage 1 (Entropy-based pruning) alone at a 70% pruning rate achieves an FID of 32.8622 at 60K iterations. Table 1 lists the results for the same 70% pruning rate at 100K iterations, where the best result after the main one is 39.34. This shows all the remaining methods perform worse than just standard entropy-based pruning.
3. The ablation study in Table 5 is not a fair comparison. For 85% pruning, the authors evaluate their method under its best configuration (Stage II on DH↑30%) according to Table 3, but do not grant competing samplers an equivalent hyperparameter tuning, which gives their method an unfair edge.
4. The paper focuses on RS generative foundation models but limits its experiments to optical (RGB) and near-infrared data. The authors explicitly state this limitation. This is a significant weakness, as a foundation model specialized remote sensing should handle the multi-modal bands. The proposed entropy-based filter, for example, may not be suitable for SAR data, which is dominated by speckle noise rather than low-information uniform color.
5. The proposed pipeline is a straightforward combination of (i) global Shannon entropy filtering and (ii) centroid‑based clustering/sampling. The core idea of deriving prototypes/centroids from curated datasets and then pull the unlabeled pool toward those prototypes resembles DINOv2/V-JEPA 2 automatic data curation pipeline. The paper does not discuss this conceptual overlap or carefully position what is new beyond swapping to RS‑specific priors.

**Questions:**

1. At 70% pruning, why does Stage I-only (FID 68.38, Table 3) outperform the full Stage I+II pipeline (FID 70.04, Table 3), while in Table 1, the full pipeline ("Ours") is used (FID 28.46)? Can you mention what configuration you used in table 1 for Stage I and II and also the number of clusters?
2. It was mentioned that you used Git-RSCLIP for feature extractor, because compared with backbones such as DINOv2, it delivers more reliable and discriminative embeddings for RS imagery. It there a source to support that claim that compares those two?
3. The sampling ablation in Table 5 appears to be biased. The "Ours" method uses its optimal input subset (DH↑30%) as identified in Table 3. Can the authors ensure this is a fair comparison? Was this DH↑30% subset also found to be optimal for the other sampling strategies, or your method simply benefiting from a best-case configuration?

---

### Official Review · Reviewer_isxz · 2025-10-31

**Soundness:** 2
**Presentation:** 2
**Contribution:** 2
**Rating:** 4
**Confidence:** 4

**Summary:**

The paper explores data pruning strategies for remote sensing generative foundation models (RS-GFMs) to reduce training cost while maintaining model performance. It introduces three pruning techniques, e.g., uncertainty-based, representation similarity-based, and gradient influence-based, to identify and discard redundant or low-value samples before large-scale generative model training. Experiments are performed using a diffusion-based RS-GFM on multispectral and hyperspectral datasets (e.g., BigEarthNet-S2, Houston HSI) to evaluate efficiency and accuracy trade-offs. The authors claim that up to 30–40% of training data can be pruned with negligible degradation in reconstruction and downstream classification performance

**Strengths:**

Motivation: The problem of excessive data redundancy in the RS foundation model pretraining is practically important, given the scale of current multimodal EO datasets.

Clarity of goal: The work explicitly focuses on data efficiency, a relevant aspect often overlooked in the current GFM literature.

Experimental variety: Multiple pruning criteria are tested, and quantitative evaluations include both reconstruction and downstream metrics.

Potential utility: If robust, such pruning could reduce computation and storage requirements in large RS model training pipelines.

**Weaknesses:**

Lack of novelty.
The three proposed strategies are direct adaptations of well-known data pruning and active learning techniques (e.g., influence functions, gradient norm, sample uncertainty from model entropy, and representation similarity from feature clustering). No novel algorithmic formulation or theoretical contribution is introduced. The work essentially repackages standard machine learning pruning heuristics for remote sensing data without domain-specific adaptation or innovation.

No significant performance gain or insight.
The results show only minor computational savings (~20–30%) with marginal accuracy drops. However, no deeper analysis is provided on why certain samples are redundant or how pruning affects spectral diversity, spatial distribution, or sensor-specific characteristics—factors crucial in RS GFMs.

Experimental limitations.

The tested models are small-scale diffusion or autoencoder variants (tens of millions of parameters), not true foundation models (hundreds of millions to billions). Thus, conclusions may not generalize.

The pruning ratios, metrics, and datasets are limited; no experiments on cross-sensor generalization, multimodal inputs, or real foundation-scale data are shown.

Ablation settings and hyperparameters (e.g., gradient computation batch size, entropy thresholds) are under-specified.

Comparisons with modern data selection or dataset distillation methods (e.g., CRAIG, GradMatch, K-center, D2 pruning, ActiveDF) are missing.

Clarity and presentation issues.
Figures are small and difficult to read, with low resolution and poor contrast; key visual comparisons (e.g., Figure 3 and Figure 5) are nearly illegible. The text contains overlapping abbreviations and inconsistent notation. The flow between Sections 3–5 is disorganized, mixing method and results.

Positioning gap.
The paper fails to connect its work to the current GFM ecosystem (e.g., SpectralGPT, EarthGPT, OmniSat, CROMA, SatMAE). It does not analyze how pruning interacts with pretraining paradigms like masked modeling or diffusion-based generation. This omission makes the contribution appear detached from the state of the art.

**Questions:**

How does pruning affect the spectral coverage and spatial distribution of the retained dataset? Are important edge cases or rare materials disproportionately removed?

How would the methods scale to real GFM settings with 100M+ parameters and global EO datasets?

Can pruning be made sensor-aware or modality-adaptive rather than using generic uncertainty or gradient scores?

Why were state-of-the-art data selection methods like GradMatch, K-center, and D2 not compared?

Can the authors provide clearer, higher-resolution figures and detailed algorithm pseudocode to improve readability and reproducibility?

---

### Official Review · Reviewer_3XeW · 2025-11-01

**Soundness:** 2
**Presentation:** 3
**Contribution:** 3
**Rating:** 4
**Confidence:** 4

**Summary:**

This paper addresses the critical challenge of training generative foundation models for Remote Sensing on large-scale, noisy, and redundant datasets. The authors propose a two-stage pruning approach that combines entropy-based filtering (Stage I) with scene-aware clustering and stratified sampling (Stage II). The key innovation is leveraging existing RS scene classification datasets as reference benchmarks to guide clustering, avoiding the computational cost of clustering large-scale unlabeled data directly. Experiments demonstrate that this pruning strategy significantly improves results over training on the full dataset.

**Strengths:**

- The paper addresses a critical problem in RS generative modeling. The authors clearly articulate the unique challenges of RS data (redundancy, noise, class imbalance) and why existing pruning methods are insufficient. The idea of using a "prior bank" of existing classification datasets to guide the clustering of a massive, unlabeled generative dataset is creative.
- The method is tested on multiple large-scale datasets, the ablation studies are comprehensive and robustly support the design choices.
- The paper is well-written and clearly structured.

**Weaknesses:**

1. The number of clusters, $K$ is a crucial and sensitive hyperparameter. Figure 3 shows that performance degrades if $K$ is too large or too small. However, the paper does not provide a principled method for selecting the optimal $K$. This seems to require an expensive hyperparameter sweep, which undermines the method's "efficiency" claim.
2. The success of Stage II relies on the careful manual curation of the "prior bank". The authors note in Figure 3 that simply aggregating more datasets can hurt performance due to cross-dataset heterogeneity. This implies a user must carefully select complementary datasets, which is a non-trivial prerequisite.
3. The paper's core contribution, Stage II, is not universally beneficial and its efficacy is questionable. As shown in Table 3, the combined Stage I+II pipeline performs worse than the entropy-only (Stage I) baseline in many scenarios (e.g., for Git-10M at 70% pruning).

**Questions:**

1. The entire Stage II pipeline (both centroid generation and sample assignment) is critically dependent on the quality and properties of the chosen feature extractor, Git-RSCLIP. Could the authors comment on how sensitive the Stage II results are to the choice of feature extractor? The paper does not investigate how this choice impacts the results.
2. Did the authors conduct multiple runs with different random seeds? Are the reported FID improvements statistically significant? What is the performance variance across different random seeds (e.g., for K-means initialization and model training)?
3. The paper states that FID is computed against a selected set of 5,000 "high-quality" images (Section 4.1). How was this test set selected? Was it a simple random sample from the full dataset, or was it filtered using criteria similar to Stage I (e.g., high entropy)? If the test set itself is "clean," this could introduce evaluation bias that unfairly favors models trained on the pruned (cleaner) data over those trained on the full, noisy dataset.

---

### Note · Authors · 2025-11-13

I have read and agree with the venue's withdrawal policy on behalf of myself and my co-authors.